# Complications and Implant Survival of Total Knee Arthroplasty in People with Hemophilia

**DOI:** 10.3390/jcm11216244

**Published:** 2022-10-23

**Authors:** Emerito Carlos Rodriguez-Merchan, Hortensia De la Corte-Rodriguez, Teresa Alvarez-Roman, Primitivo Gomez-Cardero, Carlos A. Encinas-Ullan, Victor Jimenez-Yuste

**Affiliations:** 1Department of Orthopedic Surgery, La Paz University Hospital-IdiPaz, 28046 Madrid, Spain; 2Osteoarticular Surgery Research, Hospital La Paz Institute for Health Research—IdiPAZ (La Paz University Hospital—Autonomous University of Madrid), 28046 Madrid, Spain; 3Department of Physical and Rehabilitation Medicine, La Paz University Hospital-IdiPaz, 28046 Madrid, Spain; 4Department of Hematology, La Paz University Hospital-IdiPaz, 28046 Madrid, Spain

**Keywords:** hemophilia, knee, total knee arthroplasty, complications, implant survival

## Abstract

Total knee arthroplasty (TKA) is a commonly used option in advanced stages of knee arthropathy in people with hemophilia (PWH). The objective of this article is to determine what the complication rates and implant survival rates in PWH are in the literature. A literature search was carried out in PubMed (MEDLINE), Cochrane Library, Web of Science, Embase and Google Scholar utilizing the keywords “hemophilia TKA complications” on 20 October 2022. It was found that the rate of complications after TKA in PWH is high (range 7% to 30%), although it has improved during the last two decades, possibly due to better perioperative hematologic treatment. However, prosthetic survival at 10 years has not changed substantially, being in the last 30 years approximately 80% to 90% taking as endpoint the revision for any reason. Survival at 20 years taking as endpoint the revision for any reason is 60%. It is possible that with a precise perioperative control of hemostasis in PWH, the percentage of complications after TKA can be diminished.

## 1. Introduction

Total knee arthroplasty (TKA) is a surgical intervention that frequently has to be performed in people with hemophilia (PWH) when they suffer from very painful advanced arthropathy whose pain does not subside with conservative treatment (hematologic prophylaxis, analgesics, anti-inflammatory drugs, Physical Medicine and Rehabilitation (PMR), and intra-articular injections of hyaluronic acid or platelet-rich plasma (PRP). In addition, TKA usually works well in PWH in terms of pain and improved quality of life (QoL) [1,2].

It is essential that TKA is always performed with adequate control of hemostasis by the hematologists in charge of the patient, which is accomplished by intravenous administration of clotting factor deficiency factor (F) VIII or FIX at doses deemed necessary by the hematologists and for as long as they deem appropriate [3,4,5,6].

According to Escobar et al. planning and carrying out TKA in PWH is most effective with the implication of a specialist and expert multidisciplinary team (MDT) at a hemophilia hospital. Rehabilitation after surgery must start soon, with focus on management of hemostasis and pain. Surgery in PWH and inhibitors needs even more cautious planning [7]. However, despite carrying out this kind of surgery with the support of an MDT, complications after TKA in PWH are frequent. Logically, such complications are even more frequent if TKA is performed in a center not specialized in the treatment of hemophilia [1,2].

In osteoarthritis (OA), the survival rate of 10 years for revision published in 2022 by Ueyama et al. was 98% [8]. In OA, the reported survival of 20 years for revision was 96% [9]. The reported complication rate in OA was 7% [10]. 

The hypothesis of this article is that PWH have a higher rate of complications and lower implant survival than people without hemophilia. The research questions were the following: Do PWH have more complications when implanted with a TKA than people without hemophilia? Is implant survival different in PWH than in people without hemophilia? The aim of the study is to determine whether PWH have more complications than people without hemophilia when implanted with a TKA and whether prosthetic survival is lower in PWH than in people without hemophilia.

## 2. Methods

A PubMed (MEDLINE), Cochrane Library, Web of Science, Embase and Google Scholar search of reports on complications after TKA in PWH was conducted. The key words utilized were “hemophilia TKA complications”. The main inclusion criteria were reports focused on the complications after TKA in PWH. Studies not focused on such risk factors were disregarded. The searches were dated from the creation of the search engines until 20 October 2022. From the 5138 articles (2870 in the Web of Science, 2030 in Google Scholar, 159 in Embase, 77 in PubMed (MEDLINE), 2 in The Cochrane Library), we chose those that seemed most directly related to the title of this article (53 articles).

Medical subject headings (MeSH) terms were “hemophilia”, “TKA” and “complications” and the strig was the following: “hemophilia” OR TKA OR “complications”; “hemophilia” AND (TKA OR “complications”); (“hemophilia” AND TKA) OR “complications”. Inclusion and exclusion criteria: The articles that were most directly related to the title of the article were subjectively included; the rest were excluded. Duplicate references were highlighted. To delete duplicate citations we right-clicked on any highlighted reference. Then, we selected “move references to trash”. To achieve data extraction we used the three steps of the ETL process (extraction, transformation, and loading).

This article is not a systematic literature review, but a narrative review of the literature of the articles found in the various existing databases that, according to our criteria, were considered to be closely related to the title of the article.

## 3. Results

### 3.1. Published Series

In 1989, Figgie et al. analyzed 19 TKAs carried out in PWH. The average follow-up was of 9.5 years [11]. A total of 13 knees had good or excellent outcomes, and 6 knees had poor results. Those subjects with excellent outcomes had sustained good function and pain alleviation. Four of the six poor results were found in the first seven TKAs carried out, when only 80% FVIII coverage was utilized in the perioperative phase. Since the utilization of 100% FVIII coverage began, the rate of poor results diminished. A total of 10 of the 19 knees experienced adverse events: 1 periprosthetic joint infection (PJI), 6 superficial skin necroses, 3 nerve paralyses, 7 postoperative hemorrhages, and 1 transfusion reaction. Six of the seven knees that experienced TKA under 80% factor VIII coverage suffered adverse events. Once 100% FVIII coverage was initiated, the only adverse events were one skin necrosis and three postoperative hemorrhages. The percentage of radiographic failure was high, with progressive radiolucent lines in 13 of 19 tibial components, associated with tibial component displacement in 3 knees. The aforementioned radiographic findings did not correlate with clinical outcomes. However, pain alleviation and ameliorated function were preserved at longer follow-up periods. The best outcomes were attained under 100% FVIII coverage utilizing a posterior stabilized (PS) design and patellar resurfacing [11].

In 1990, Kjaersgaard-Andersen et al. analyzed 13 semiconstrained TKAs carried out in 9 males with hemophilia A with a mean age of 38 years [12]. The average FVIII amount during hospital stay was 84,222 units, and the average hospitalization period was 33 days. Four patients (44%) died during the study period, three from acquired immunodeficiency syndrome (AIDS) and one from sudden cardiac arrest. At the time of TKA, one of the subjects who died from AIDS had a positive test for human immunodeficiency virus (HIV). He died 3 months after surgery. The other two patients acquired AIDS 1 year and 4 years after surgery. The average follow-up period was 43 months. Utilizing the Hospital for Special Surgery (HSS) knee rating scale, the results were excellent in nine knees and good in three knees. All subjects were fully alleviated of pain. TKA in hemophiliacs seemed to be an efficacious treatment for hemophilic arthropathy [12].

In 2003, Legroux-Gérot et al. assessed the outcomes of 17 TKAs (12 patients) and its influence on both QoL and clotting factor utilization [13]. The TKAs were carried out between 1986 and 1996, and the mean follow-up was 54 months. Mean age at the time of TKS was 39 years. QoL was assessed utilizing the Short Form 36 (SF-36). In 94% of the subjects the outcomes were good or excellent. The amelioration was greatest for pain. Recurrent joint hemorrhages in six subjects and development of an inhibitor in two subjects were the only adverse events during the postoperative period. Clotting factor utilization did not diminish substantially after TKA. Legroux-Gérot et al. expressed that TKA for hemophilic arthropathy provided good outcomes that translated into QoL gains [13].

In 2004, Sheth et al. reported the outcomes of 14 TKAs in 9 PWH utilizing posterior cruciate ligament (PCL)-sacrificing designs [14]. The mean follow-up in surviving subjects (13 knees) was 77 months. Pain, functional score, flexion deformity, and flexion range improved significantly. Nine adverse events happened in 6 knees. One subject died from HIV-related adverse events. No subject seroconverted to HIV during the follow-up time [14].

The outcomes of 24 modular Genesis II (Smith & Nephew, Memphis, TN, USA) TKAs, carried out in 20 subjects (mean age, 36 years) with hemophilia, were prospectively reviewed in 2007 by Innocenti et al. [15]. The mean follow-up was 4.4 years. Knee score, the mean knee flexion contracture and the mean total flexion arc ameliorated. The outcomes of this report showed that the use of modular design improved the functional outcomes of TKA in hemophilic arthropathy, which led to a better range of motion (ROM) and lower flexion contracture [15].

In 2008, Chiang et al. assessed the clinical and functional results of TKA and reasons of prosthetic failure in 26 PWH (35 TKAs) [16]. The mean age was 34.2 years and the mean follow-up was 82.2 months. Three subjects required manipulation under anesthesia (MUA) because of an inadequate ROM. Three infections were managed with debridement and one of them experienced knee fusion after removal of prosthesis. Two subjects experienced revision TKA. One of them was due to loosening of the femoral component. The other one experienced revision TKA due to wear of the insert. Chiang et al. stated that TKA seemed to be an effective procedure to accomplish pain alleviation and to improve function in PWH [16].

In 2009, Massin et al. published the outcomes of 128 TKAs from 5 specialized centers. Only knees with preoperative flexion less than 90° were included [17]. Adverse events were 3 skin necroses, 2 infections, 2 femoral fractures, and 1 sciatic nerve paralysis. TKA rendered substantial flexion improvement. It frequently needed tibial tuberosity osteotomy to ameliorate exposure and preclude injury to the extensor mechanism [17].

In 2012, Feng et al. assessed the results of 25 TKAs (19 subjects) [18]. Average patella thickness was 16.3 mm and all patellas were managed by patelloplasty. The subjects were followed for an average postoperative period of 41 months. The preoperative HSS (Hospital for Special Surgery) score was 51 on average. The postoperative HSS score was 91. ROM was modified to 82°, compared with 55° preoperatively. A total of 13 subjects with flexion contracture were corrected from 19° to 2.7°. Four subjects complained of mild but sustainable pain in the anterior part of the knee joint [18].

In 2014, Westberg et al. studied clinical results and adverse events of 107 TKAs in 74 hemophilic subjects with special focus on prosthetic survival and PJI [19]. Mean follow-up was 11.2 years. Percentages of survival at 5 years and 10 years, with component removal for any cause as the end point, were 92% and 88%, respectively. A total of 28 TKAs were removed after 10 years on average. The most common reason of failure was aseptic loosening (14 knees) and PJI (seven knees). The infection rate was 6.5%. A painless knee was found in 93% of the TKAs at the final follow-up. The medium and long-run outcomes of primary TKA showed good prosthetic survival at 5 and 10 years with an excellent alleviation of pain. PJI was still a primary concern compared to patients without hemophilia [19].

In 2015, Strauss et al. assessed the clinical results of 23 TKAs carried out in 21 hemophilic subjects with preoperative ROM of 50° or less [20]. Mean follow-up was 8.3 years. There were one late PJI, and one aseptic implant loosening (8.7% complication rate). Nine subjects who required VY-quadricepsplasty for knee exposure suffered a mean postoperative extensor lag of 7°. Strauss et al. affirmed that although the clinical result was inferior compared to nonstiff knees previously published, TKA can be successfully carried out in PWH [20].

In 2015, Rodriguez-Merchan analyzed 74 PWH treated with TKA (N = 88) over a period of 13 years [21]. The same kind of design was utilized in all cases. A total of 14 subjects had 2-stage bilateral TKAs. The mean subject age was 38.2 years. A total of 14 subjects were positive for HIV and 32 for hepatitis C virus (HCV). The mean follow-up was 8 years. The percentage of prosthetic survival with implant removal for any cause regarded as final endpoint was 92%. Reasons of TKA failure were PJI (6.8%) and aseptic loosening (2.2%). Clinical outcomes of the primary TKAs in this report showed good prosthetic survival and excellent pain alleviation [21].

In 2017, Ernstbrunner et al. provided clinical and radiological long-run outcomes of 43 TKAs implanted in 30 PWH (study with level 4 of evidence) [22]. After a mean of 18 years, 15 subjects (21 knees) with a mean age of 58 years were accessible for follow-up. In 13 (30%) of the 43 knees, revision arthroplasty was required due to PJI or aseptic loosening among which 8 (19%) were due to aseptic loosening and 5 (12%) were due to hematogenous infection. The estimated survival percentages of 20 years, with revision for any cause or infection as the end points, were 59 and 82%, respectively. A total of 86% rated their outcome as either good or excellent. TKA in hemophilic subjects was associated with high revision, loosening and infection percentages after 18 years [22].

In 2017, Szmyd et al. analyzed 40 TKAs in 35 subjects. The mean follow-up was 19.4 months. The mean age of subjects was 36.7 years. The pain intensity was considerably diminished 12 months after the surgery. A significant improvement in patients’ functioning was recorded. TKA seemed to be a very good therapy for subjects with advanced hemophilic arthropathy. TKA considerably diminished pain intensity. TKA markedly improved subjects’ functioning in daily life. Subjects were very satisfied with the results of TKA [23].

In 2018, Song et al. assessed mid-run results and adverse events of TKA in hemophilic arthropathy [24]. They retrospectively reviewed 131 primary TKAs. The mean age was 41 years, and the mean follow-up was 6.8 years. Adverse events happened in 17 knees (13%): 7 articular hemorrhages, 4 periprosthetic fractures, 3 PJIs, 2 stiffness, and 1 medial collateral ligament injury. The mid-run outcomes of TKA in PWH were satisfactory in pain alleviation, ameliorated function and diminished flexion contracture [24].

A total of 18 TKAs carried out in 15 subjects with hemophilia during a period of 24 years were analyzed in 2019 by Santos Silva et al. Mean follow-up was 11.3 years. The survival percentage of 10 years with TKA removal as end point was 94.3%. Only two subjects needed perioperative transfusion. The rate of postoperative adverse events was 27.8% (two PJIs, two knee stiffness, and one case of recurrent articular hemorrhage) [25].

In 2022, Oyarzun et al. analyzed 41 TKAs (19 cases were bilateral) [26]. Six patients needed revision (6.66%) due to PJI. The percentage of TKA survival at 5 years was 92% (range 82–96%) [26].

In 2020, Bae et al. analyzed a series of 78 TKAs in 56 PWH [27]. The mean age was 38.7 years old and the mean follow up was 10.2 years. Postoperative adverse events happened in 12 knees (15.4%). The percentage of hospital readmission in the 30 days after discharge was 6.4%. Revision TKA was carried out in 3 knees for PJI and in 1 knee for loosening of the tibial component loosening. The prosthesis survival rates at 10 years and 13 years were 97.1% and 93.2%, respectively [27].

In 2022, Kleiboer et al. analyzed 98 TKAs in PWH of which 25% were complicated by major bleeding [28]. The risk of major bleeding was augmented by the presence of an inhibitor, increased body mass index (BMI), and non-use of an antifibrinolytic medication. Neither continuous clotting factor infusion (versus bolus infusion) nor pharmacologic thromboprophylaxis were associated with bleeding risk. Use of antifibrinolytic medications was associated with diminished risk [28].

In 2022, Wang et al. assessed the mid-run results of 32 TKAs (28 patients) for end-stage hemophilic arthropathy [29]. The follow-up was 69.1 months. Significant differences between the preoperative and final follow-up values of flexion contracture, ROM, clinical KSS (Knee Society Score), functional KSS, and VAS (Visual Analog score) were found. The incidence of adverse events was 15.6% and the rate of satisfaction was 100% [29].

In a population-based study published in 2022, Chen et al. analyzed 103 primary TKAs (75 subjects). Unilateral TKA was carried out on 47 subjects and bilateral TKAs on the remaining 28 subjects, including 12 simultaneous and 16 staged surgeries. The mean age of patients was 32.3 years, and the mean follow-up was 77.9 months. Failure occurred in eight subjects (8.5%) at mean 32.8 months after surgery. Four subjects suffered aseptic loosening and four experienced PJI. The prosthesis survivorship of 10 years was 88.6%. For subjects experiencing unilateral TKA, the mean length of stay (LOS) was 14 days. The prosthesis survivorship of 10 years was 88.6%. [30].

Table 1 summarizes the main series reported on TKA in PWH. Figure 1 shows the rate of complications reported between 1989 and 2022. Figure 2 depicts the rates of postoperative bleeding after TKA in PWH. Figure 3 shows the rates of PJI after TKA in PWH. Figure 4 depicts the survival rates of TKA in PWH.

### 3.2. Venous Thromboembolism (VTE)

In 2019, Peng et al. reported a 1.5% prevalence of clinically significant VTE in PWH experiencing TKA without chemoprophylaxis and a modified coagulation factor substitution [31]. They affirmed that given the low incidence of clinically significant VTE in their study, routine chemoprophylaxis in PWH experiencing TKA might not be required [31].

### 3.3. Arterial Pseudoaneurysms

An arterial pseudoaneurysm must be suspected when after the surgical procedure there is severe bleeding that does not cease with adequate management with intravenous injection of the deficient coagulation factor [32,33]. The diagnosis and treatment of a pseudoaneurysm have to be done quickly to prevent complications. The diagnosis must be confirmed by duplex ultrasonography (US), standard angiogram, computed tomography (CT) angiogram or magnetic resonance angiography. There are several options for the treatment of pseudoaneurysms. Small pseudoaneurysms can be solved with conservative noninterventional treatment. It includes outside pressing, US probe pressing or US-guided thrombin injections. In bigger pseudoaneurysms, endovascular techniques, such as coil embolization, are now favored. If the aforementioned techniques are futile, standard surgical treatment with simple ligation or arterial reconstruction have to be performed [32,33].

### 3.4. Case Reports

#### 3.4.1. Intraoperative Popliteal Artery Injury

In 2017, Feng et al. reported the case of a 48-year-old male subject with severe hemophilia A and stiff knees that experienced bilateral TKAs [34]. Left popliteal artery injury was detected at the end of the left TKA. Urgent angiography confirmed the diagnosis of the left popliteal artery transection. With clotting FVIII replacement treatment, open repair was carried out by end-to-end vascular bypass with the autograft of the large saphenous vein. Left lower limb was reperfused 4 h after the beginning of the ischemia. The subject recovered uneventfully. Postoperative Doppler examination demonstrated the left popliteal artery remained patent [34].

#### 3.4.2. Brucella Infection

In 2017, Mortazavi et al. reported a 28-year-old man with *Brucella* infection of TKA, who at first experienced conservative management but then required a two-stage revision TKA [35].

#### 3.4.3. Postoperative Flexion Contracture

In 2021, Liawrungrueang et al. reported the successful treatment of flexion contracture after primary TKA in a 20-year-old-man with hemophilia A by open soft tissue contracture releasing and serial casting [36]. Table 2 summarizes the complications of TKA in PWH.

### 3.5. Is Drain after TKA Necessary?

In a prospective study, Haghpanah et al. compared the outcomes of drain protocol (42 TKAs in 39 subjects, mean age 35.5 years) with no-drain-protocol (38 TKAs in 27 subjects, mean age 35.7 years) [37]. Patients were followed for at least 1 year. There was no statistical difference between the two groups in terms of knee scores, blood loss, postoperative pain, fever, time to regain the ROM and infection. Two subjects in the drain group and one subject in the no drain group were reoperated due to PJI. No subjects required blood transfusion in each group [37]. In a prospective randomized clinical trial, 176 subjects with hemophilia who experienced TKA were analyzed by Mortazavi et al. [38]. The study group consisted of 88 subjects (108 knees) in which we did not insert suction drain and the control group included 88 subjects (106 knees) in which drain was inserted at the end of the surgery. No differences in the mean VAS value between both groups were observed. Mortazavi et al. concluded that there was no basis for the utilization of drain after primary TKA in PWH [38].

### 3.6. Special Scenarios

#### 3.6.1. Computer-Navigated TKA

In 2013, Cho et al. assessed the outcomes of 27 computer-navigated TKAs in 25 patients with hemophilic arthropathy [39]. The clinical results were substantially ameliorated after the surgery. There were no adverse events specific to the computer-navigated TKA [39].

#### 3.6.2. Robot-Assisted TKA

In 2016, Kim et al. assessed 32 robot-assisted TKAs in 29 hemophilia subjects [40]. The mean follow-up period was 5 years. Adverse events included early hematoma in three knees, heterotopic ossification in three knees, and PJI in two knees [40].

#### 3.6.3. TKA in Patients with Inhibitors

In 2021 Carulli et al. assessed the results of 18 hemophilic patients with inhibitors (26 TKAs) [41]. Subjects were divided in two groups: group A (primary total TKA): 13 subjects underwent 19 TKAs; and B (revision): 5 subjects underwent 3 revision TKAs. All subjects received the same hematological prophylaxis (recombinant factor VII activated-rFVIIa). The median follow-up was 12.2 years for group A, 8.6 years for group B. Few adverse events were found; the survival rate was 94.7% at 15 years. All patients reported satisfaction, pain alleviation and ameliorated functional ability. The use of continuous infusion of rFVIIa showed an appropriate hemostatic effect and low percentage of complications. Revision TKAs were more susceptible to complications compared to primary TKA [41].

#### 3.6.4. Simultaneous Bilateral TKA

A study showed that bilateral TKA was a safe and cost-effective procedure for hemophilic arthropathy with similar medium-run outcomes compared to unilateral TKA [42].

#### 3.6.5. Medicare Beneficiaries with a Diagnosis of Hemophilia

A study found that adverse medical events were more common among PWH: postoperative bleeding, deep vein thrombosis (DVT), pulmonary embolism and blood transfusions. PWH showed higher odds of PJI (1.78 versus 0.98%). Reimbursements of 90 days were higher for subjects with hemophilia (mean: $22,249 versus $13,017) [43].

#### 3.6.6. TKA for a Stiff Knee of PWH

In a study, 67 primary TKAs for PWH (mean age, 48 years) were carried out, and incisional approaches to joint were standard (58 cases) and V-Y quadricepsplasty (V-Y) (9 cases). Preoperative ROM and flexion were significantly associated with V-Y. Ono et al. affirmed that primary TKA for PWH utilizing a standard approach might be carried out before the phase preoperative flexion < 45° and ROM < 35° [44].

### 3.7. Comparative Studies

#### 3.7.1. Continuous Infusion versus Bolus Injection

In 2017 Park et al. assessed the efficacy of continuous infusion (CI) of coagulation factor concentrates during the perioperative period compared to bolus infusion (BI) [45]. A total of 42 TKAs were carried out in 31 subjects with severe hemophilia A. Although good control of hemostasis was accomplished utilizing either method during the perioperative period of TKA, CI appeared to be more tolerable and effective than BI to provide perioperative blood management in PWH experiencing TKA [45].

#### 3.7.2. Hemophilia versus Non-Hemophilia

In 2011 Sikkema et al. compared the outcomes of TKA in subjects with and without hemophilia retrospectively [46]. The adverse events and long-run outcomes of 21 TKAs carried out in 22 hemophilia subjects were compared with those of 42 TKAs in subjects without hemophilia. Subjects were matched for gender, year of surgery and age. Hemarthrosis happened in 52% of the TKAs carried out in the hemophilia subjects, while hemarthrosis occurred in 7% of the TKAs of the control group. In the hemophilia group, the rate of PJI was 7%, while it was 13% in the control group. Subjective function was good in 76% of TKAs in hemophilia subjects versus 71% in TKAs in controls [46].

In 2019 Wang et al. assessed the risk of complications of PWH who experienced TKA utilizing information from the National Health Insurance Research Database [47]. PWH had longer LOS and greater total hospital expenses compared to people without hemophilia. There were no differences between the rates of adverse events of 30 days and 90 days, a PJI of 1 year, reoperation and mortality between PWH and people without hemophilia [47].

#### 3.7.3. Hemophilia versus Osteoarthritis and Rheumatoid Arthritis

In 2020 Li et al. analyzed the adverse events of 2083 TKA in 1515 PWH compared with osteoarthritis (OA) and rheumatoid arthritis (RA) [10]. The overall rate of adverse events in the hemophilic arthropathy group was 21.79%, which was much greater than the OA or RA group (7.08% and 8.70%, respectively). The main adverse events were loosening of the implant and wound dehiscence. For PWH, more adverse events happened in the period more than 1 year after TKA, when compared with OA (33.33% vs. 11.43%). Among the potential risk factors, subjects with hemophilia B and severe hemophilia had substantially greater percentages of adverse events [10].

### 3.8. Meta-Analysis

In 2016 Moore et al. published a meta-analysis of 20 studies (336 TKAs in 254 PWH). The mean follow-up was 6.3 years. Statistically significant ROM improvements were encountered. Knee scores showed statistically significant improvements. A 31.5% rate of adverse events was found [48].

## 4. Discussion

TKA is contemplated as the management of choice for end stage arthropathy in PWH. Compared to other subjects experiencing TKA, PWH have specific characteristics, such as a bleeding trend, younger age, preoperative limited ROM, altered anatomy, and increased adverse events [1]. The use of a multimodal blood loss prevention approach that includes intra-articular tranexamic acid (TXA) (MBLPM-TXA) in PWH who experience TKA is effective in reducing the percentages of transfusion [49].

Perioperative treatment with expert orthopedic and hematological counsel is advised to optimize results in PWH [50]. TKA (both primary and revision) should be carried out in hospitals specialized in orthopedic surgery, physical and rehabilitation medicine, and hematology [2].

A coagulation factor level <93.5% or hematocrit level of <38.2% might be a substantial risk factor for increasing perioperative blood loss [51]. Adequate intra- and postoperative care to avert postoperative residual contracture is needed in PWH [52]. In the last 30 years, there has been a decrease in the rate of TKA, probably indicating the impact of extensive utilization of tertiary hematological prophylaxis [53].

It seems logical to think that inadequate hemostasis at the time of surgery will not only cause the patient to bleed more in the postoperative period but will also increase the risk of prosthetic infection (since blood is an excellent breeding ground for bacteria). Such infection can appear more or less early, often leading to septic loosening of the implant. In fact, it is well known that the average infection rate after TKA in PWH is 7% (compared to 1–2% in people without hemophilia). This would suggest that the percentage of prostheses still in situ in the long term is probably lower in PWH than in persons without hemophilia, since an average of 7 out of 100 prostheses in PWH can loosen due to infection versus 1–2 out of 100 in persons without hemophilia. In other words, although there is no irrefutable evidence that inadequate hemostasis during surgery is the cause of late loosening of the implant, in our opinion it seems a logical assumption that will undoubtedly have to be confirmed in the future with clinical evidence.

The main limitations of this article are that although the major databases (PubMed, Cochrane Library, Embase, Web of Science, and Google Scholar) have been used, it is not a systematic review of the literature but a narrative review of the literature. Furthermore, the selection of articles was carried out subjectively, considering only those closely related to the title of the article. Therefore, it is possible that some important articles on the topic were not included.

## 5. Conclusions

The rate of adverse events after TKA in PWH remains high (range 7% to 30%), although it has improved during the last two decades, possibly due to better perioperative hematologic treatment. However, prosthetic survival at 10 years has not changed substantially, being in the last 30 years approximately 80% to 90% taking as an endpoint the revision for any reason. Survival at 20 years taking as an endpoint the revision for any reason is 60%. It is possible that with an exquisite perioperative control of hemostasis in PWH, the percentage of complications after TKA can be reduced.

## Figures and Tables

**Figure 1 jcm-11-06244-f001:**
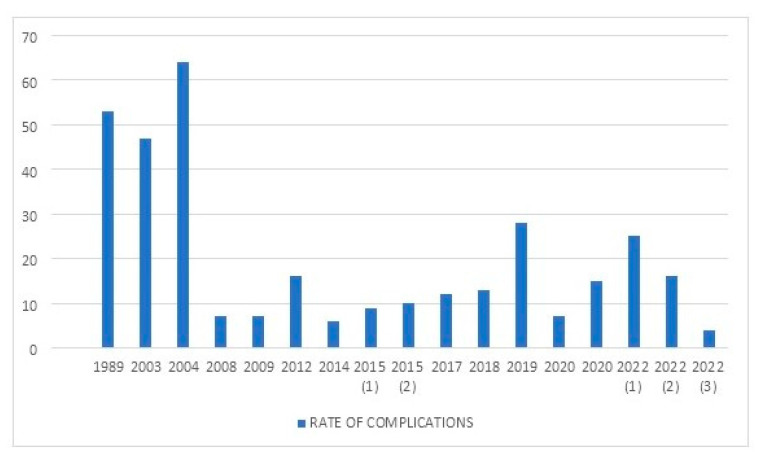
Rates of complications of total knee arthroplasty (TKA) in people with hemophilia (PWH) from 1989 to 2022. In 2015 there were two articles ((1) and (2)), while in 2022 there were three articles ((1)–(3)).

**Figure 2 jcm-11-06244-f002:**
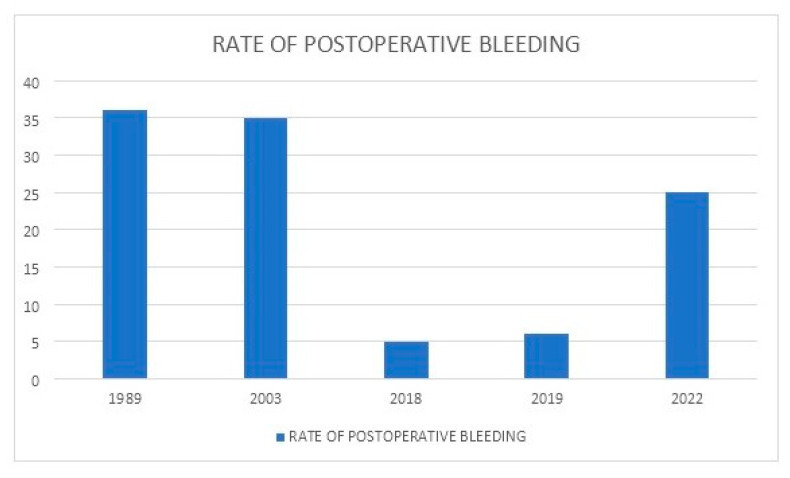
Rates of postoperative bleeding after total knee arthroplasty (TKA) in people with hemophilia (PWH) from 1989 to 2022.

**Figure 3 jcm-11-06244-f003:**
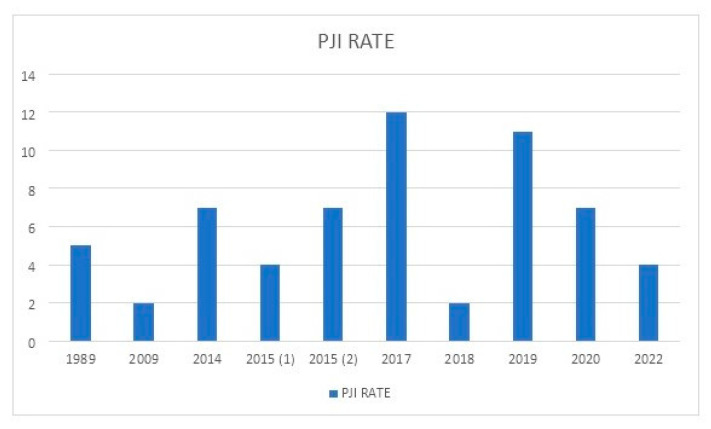
Rates of periprosthetic joint infection (PJI) after total knee arthroplasty (TKA) in people with hemophilia (PWH) from 1989 to 2022. In 2015 there were two articles [(1) and (2)].

**Figure 4 jcm-11-06244-f004:**
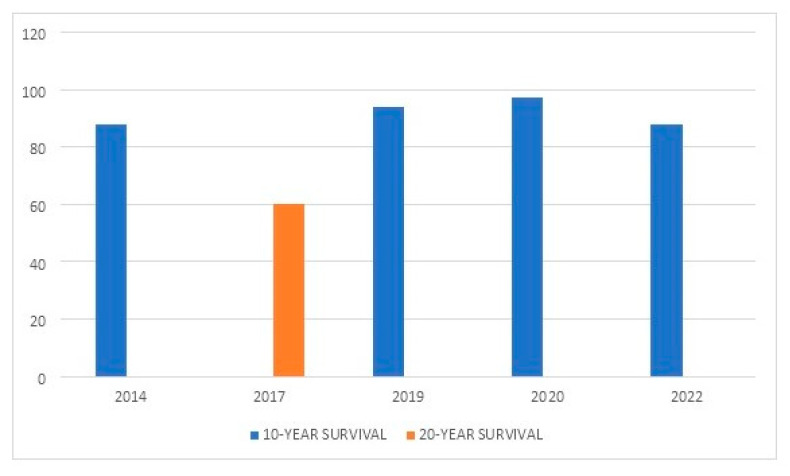
10-year-, 13-year- and 20-year-rates of implant survival of total knee arthroplasty (TKA) in people with hemophilia (PWH) from 2014 to 2022.

**Table 1 jcm-11-06244-t001:** Series of total knee arthroplasty (TKA) in people with hemophilia (PWH) published in the literature.

Authors [Reference]	Year	Number of TKAs/Patients	Average Age (Years)	Average Follow-Up	Complications	TKA Survival Rate
Figgie et al. [11]	1989	19/NA	NA	9.5 years	RATE OF COMPLICATIONS: 52,6% (10/19: 1 PJI, 6 superficial skin necroses, 3 nerve palsies, 7 postoperative bleedings, 1 transfusion reaction).	NA
Kjaersgaard-Andersen et al. [12]	1990	13/19	38	3.6 years	NA	NA
Legroux-Gérot et al. [13]	2003	17/12	39	4.5 years	RATE OF COMPLICATIONS: 47% (recurrent hemarthrosis in 6 patients and development of an anticoagulant in 2 patients).	NA
Sheth et al. [14]	2004	14/9	NA	6.4 years	RATE OF COMPLICATIONS: 64.3% (9/14).	NA
Innocenti et al. [15]	2007	24/20	36	4.4 years	NA	NA
Chiang et al. [16]	2008	35/26	34.2	6.8 years	RATE OF COMPLICATIONS: 7% (3 patients underwent MUA because of an inadequate ROM; 3 PJIs).	NA
Massin et al. [17]	2009	128/NA	NA	NA	RATE OF COMPLICATIONS: 7% (3 skin necroses, 2 PJIs, 2 femoral fractures, 1 rupture of patellar tendon, 1 sciatic nerve palsy).	NA
Feng et al. [18]	2012	25/19	NA	3.4 years	RATE OF COMPLICATIONS: 16% (4 patients complained mild but endurable anterior knee pain).	NA
Westberg et al. [19]	2014	107/74	NA	11.2 years	RATE OF COMPLICATIONS: 6.5% (PJI in 7 knees).	FIVE-YEAR: 92%. TEN-YEAR: 88%. With component removal for any reason as the end point.
Strauss et al. [20]	2015	23/21	NA	8.3 years	RATE OF COMPLICATIONS: 8.7% (1 late PJI, and 1 aseptic implant loosening; 9 patients who required VY-quadricepsplasty for knee exposure developed a mean postoperative extensor lag of 7°).	NA
Rodriguez-Merchan [21]	2015	88/74	38.2	8 years	RATE OF COMPLICATIONS: 10% (PJI 6.8%; aseptic loosening 2.2%).	92% (with implant removal for any reason regarded as final endpoint).
Ernstbrunner et al. [22]	2017	43/30	58	18 years	RATE OF COMPLICATIONS: 12% (5 hematogenous PJIs).	20-YEAR: 59% (with revision for any reason as the endpoint). 20-YEAR: 82% (with infection as the endpoint).
Szmyd et al. [23]	2017	40/35	36.7	1.6 years	NA	NA
Song et al. [24]	2018	131/NA	41	6.8 years	RATE OF COMPLICATIONS: 13% (7 hemarthroses, 1 medial collateral ligament injury, 2 stiffness, 3 PJIs, 4 periprosthetic fractures).	NA
Santos Silva et al. [25]	2019	18/15	NA	11.3 years	RATE OF COMPLICATIONS: 27.8% (2 PJIs, 2 prosthesis stiffness, 1 recurrent hemarthrosis).	10-YEAR: 94.3% (with prosthesis removal as end point).
Oyarzun et al. [26]	2020	41/22	NA	NS	RATE OF COMPLICATIONS: 6.6% (6 PJIs).	5-YEAR: 92%
Bae et al. [27]	2020	78/56	38.7	10.2 years	RATE OF COMPLICATIONS: 15.4%.	10-YEAR: 97.1%. 13-YEAR: 93.2%.
Kleiboer et al. [28]	2022	98/NA	NA	NA	RATE OF COMPLICATIONS: 25% (major bleeding).	NA
Wang et al. [29]	2022	32/28	NA	5.1 years	RATE OF COMPLICATIONS: 15.6%.	NA
Chen et al. [30]	2022	103/75	32.3	6.5 years	RATE OF COMPLICATIONS: 3.9% (PJI).	10-YEAR: 88.6%.

NA = Not available; PJI = Periprosthetic joint infection; MUA = Mobilization under anesthesia; ROM = Range of motion.

**Table 2 jcm-11-06244-t002:** Complications of total knee arthroplasty (TKA) in people with hemophilia (PWH).

Complications
Postoperative bleeding
Early hematoma
Arterial pseudoaneurysm
Periprosthetic joint infection (PJI)
Superficial skin necrosis
Inadequate range of motion (ROM): postoperative extensor lag, stiffness, postoperative flexion contracture
Periprosthetic fracture
Nerve palsy
Popliteal artery injury
Rupture of the patellar tendon
Medial collateral ligament (MCL) injury
Heterotopic ossification
Deep vein thrombosis (DVT)
Pulmonary embolism
Transfusion reaction
Acquired immunodeficiency syndrome (AIDS)
Development of an inhibitor
Mild but endurable anterior knee pain
Aseptic loosening

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
