# Peer review of "Complications and Implant Survival of Total Knee Arthroplasty in People with Hemophilia"

_jcm, 2022, doi:10.3390/jcm11216244_

Round 1

Reviewer 1 Report

What is the basis for concluding that hemostasis at surgery is a probably cause for implant loosening when most occur years after surgery? Please document the possible connection with  data.

Hemophilia is a disease that may alter several elements of the clotting mechanisms in humans and the ability to control the biological defect at the time of surgery is assumed to have relevance to late complications such as loosening or repeated hemarthrosis with periarticular fibrosis. The authors have not shown that hemostasis at surgery is a probable cause for late loosening of implants and need to restate that as a supposition not yet supported by clinical evidence.

Author Response

REVIEWER-1

We thank Reviewer-1 for his important comments. In BOLD BLACK you can see our responses to their comments, while in BOLD RED you can see sentences that we have included in REVISION-1 of the article.

What is the basis for concluding that hemostasis at surgery is a probably cause for implant loosening when most occur years after surgery? Please document the possible connection with data.

AUTHORS:

We have included the following paragraph in the "Discussion".

It seems logical to think that inadequate hemostasis at the time of surgery will not only cause the patient to bleed more in the postoperative period, but will also increase the risk of prosthetic infection (since blood is an excellent breeding ground for bacteria). Such infection can appear more or less early, often leading to septic loosening of the implant. In fact, it is well known that the average infection rate after TKA in PWH is 7% (compared to 1-2% in people without hemophilia). This would suggest that the percentage of prostheses still in situ in the long term is probably lower in PWH than in persons without hemophilia, since an average of 7 out of 100 prostheses in PWH can loosen due to infection versus 1-2 out of 100 in persons without hemophilia. In other words, although there is no irrefutable evidence that inadequate hemostasis during surgery is the cause of late loosening of the implant, in our opinion it seems a logical assumption that will undoubtedly have to be confirmed in the future with clinical evidence.

Hemophilia is a disease that may alter several elements of the clotting mechanisms in humans and the ability to control the biological defect at the time of surgery is assumed to have relevance to late complications such as loosening or repeated hemarthrosis with periarticular fibrosis. The authors have not shown that hemostasis at surgery is a probable cause for late loosening of implants and need to restate that as a supposition not yet supported by clinical evidence.

AUTHORS:

We believe that the above paragraph answers this question.

Reviewer 2 Report

The effort of authors is commendable but few major critical issue with the manuscript

1. Authors unable to draw the hypothesis and research question, so aim of study not clear and they are silent about to achieve the objective of study. why they have taken it as systematic literature review

2. search strategy is not well defined, how it is conducted, why major database left like web of science, Embase google scholar etc.

3. meSH term and search string not defined

4. Inclusion and exclusion criteria missing

5. How they deal with duplicate citations

6. how they achieve data extraction

7. Result section too long and discussion is too short

Author Response

REVIEWER-2

The effort of authors is commendable but few major critical issues with the manuscript

We thank Reviewer-1 for his important comments. In BOLD BLACK you can see our responses to their comments, while in BOLD BLUE you can see sentences that we have included in REVISION-1 of the article.

  1. Authors unable to draw the hypothesis and research question, so aim of study not clear and they are silent about to achieve the objective of study. Why they have taken it as systematic literature review

AUTHORS:

The hypothesis of this article is that people with hemophilia (PWH) have a higher rate of complications and lower implant survival than people without hemophilia.

The research questions were the following: Do PWH suffer more complications when implanted with a TKA than people without hemophilia? Is implant survival different in PWH than in people without hemophilia?

The aim of the study is to determine whether PWH have more complications than people without hemophilia when implanted with a TKA, and whether prosthetic survival is lower in PWH than in people without hemophilia.

In order to achieve the objective of this study, a literature search was performed as defined in point 2 below.

This article is not a systematic literature review, but a narrative review of the literature of the articles found in the various existing databases that, according to our criteria, were considered to be closely related to the title of the article.

  1. Search strategy is not well defined, how it is conducted, why major database left like web of science, Embase, google scholar etc.

AUTHORS:

We have changed “Methods” as follows (and deleted Figure 1):

A PubMed (MEDLINE), Cochrane Library, Web of Science, Embase and Google Scholar search of reports on complications after TKA in PWH was conducted. The key words utilized were “hemophilia TKA complications”. The main inclusion criteria were reports focused on the complications after TKA in PWH. Studies not focused on such risk factors were disregarded. The searches were dated from the creation of the search engines until 20 October 2022. From the 5138 articles (2870 in the Web of Science, 2030 in Google Scholar, 159 in Embase, 77 in PubMed, 2 in The Cochrane Library), we chose those that seemed most directly related to the title of this article (53 articles).

  1. meSH term and search string not defined

AUTHORS:

Medical subject headings (MeSH) terms were “hemophilia”, “TKA” and “complications” and the strig was the following: hemophilia” OR TKA OR “complications”; “hemophilia AND (TKA OR complications”); (“hemophilia” AND TKA) OR complications

  1. Inclusion and exclusion criteria missing

AUTHORS:

Inclusion and exclusion criteria: The articles most directly related to the title of the article were subjectively included; the rest were excluded.

  1. How they deal with duplicate citations

AUTHORS:

Duplicate references were highlighted. To delete duplicate citations we right-clicked on any highlighted reference and selected Move references to trash.

  1. How they achieve data extraction

AUTHORS:

To achieve data extraction we used the three steps of the ETL process (extraction, transformation, and loading).

  1. Result section too long and discussion is too short

AUTHORS:

We have shortened “Result” section and expanded ·Discussion” section.

We have also modified the paragraph on the limitations of the article by stating the following:

The main limitations of this article are that although the major databases (PubMed, Cochrane Library, Embase, Web of Science, and Google Scholar) have been used, it is not a systematic review of the literature but a narrative review of the literature. Furthermore, the selection of articles was carried out subjectively, considering only those closely related to the title of the article. Therefore, it is possible that some important articles on the topic were not included.

Round 2

Reviewer 1 Report

Please change your abstract to align with the final line in the conclusion about operative hemostasis

Author Response

REVIEWER-1

Please change your abstract to align with the final line in the conclusion about operative hemostasis

AUTHORS: In order to align the “Abstract” with the final line in the “Conclusion” we have changed the “Abstract” as follows (IN RED):

Abstract: Total knee arthroplasty (TKA) is a commonly used option in advanced stages of arthropathy in people with hemophilia (PWH). The objective of this article is to determine what the complication rates and implant survival rates in PWH are in the literature. A literature search was carried out in PubMed (MEDLINE), Cochrane Library, Web of Science, Embase and Google Scholar utilizing the keywords "hemophilia TKA complications" on October 20, 2022. It was found that the rate of complications after TKA in PWH is high (range, 7% to 30%), although it has improved during the last two decades, possibly due to better perioperative hematologic treatment. However, prosthetic survival at 10 years has not changed substantially, being in the last 30 years about 80% to 90% taking as endpoint the revision for any reason. Survival at 20 years taking as endpoint the revision for any reason is 60%. It is possible that with a precise perioperative control of hemostasis in PWH the percentage of complications after TKA can be diminished.

Reviewer 2 Report

Authors did required changes .

Author Response

REVIEWER-2

Authors did required changes.

AUTHORS: No changes have been made
